# Hsa_circ_0092856 Promoted the Proliferation, Migration, and Invasion of NSCLC Cells by Up-Regulating the Expression of eIF3a

**DOI:** 10.3390/biomedicines12010247

**Published:** 2024-01-22

**Authors:** Fuqiang Yuan, Masha Huang, Hanxue Huang, Xiaoyuan Mao, Pan Xie, Xi Li, Yang Gao, Feiyue Zeng, Zhaoqian Liu

**Affiliations:** 1Hunan Key Laboratory of Pharmacogenetics, Department of Clinical Pharmacology, National Clinical Research Center for Geriatric Disorders, Xiangya Hospital, Central South University, Changsha 410008, China; yuanfuqiang97@126.com (F.Y.); martha0126@163.com (M.H.); huang_hanxue@126.com (H.H.); xiaoyuanm@csu.edu.cn (X.M.); xiepan1993@126.com (P.X.); bayern@csu.edu.cn (X.L.); 2Institute of Clinical Pharmacology, Engineering Research Center for Applied Technology of Pharmacogenomics of Ministry of Education, Central South University, Changsha 410078, China; 3Department of Biochemistry and Molecular Cell Biology, College of Basic Medical Sciences, School of Medicine, Shanghai Jiao Tong University, Shanghai 200025, China; 4Department of Thoracic Surgery, Xiangya Hospital, Central South University, Changsha 410008, China; dr.gao@csu.edu.cn; 5Department of Radiology, Xiangya Hospital, Central South University, Changsha 410008, China

**Keywords:** circRNA, hsa_circ_0092856, eIF3a, NSCLC, progression

## Abstract

Circular RNA (circRNA) plays a very important regulatory role in a variety of human malignancies such as non-small-cell lung cancer (NSCLC). In the current study, we explored the role of hsa_circ_0092856 in the progression of NSCLC. We screened CircRNA from the *eIF3a* gene in the Circbase database. The biological functions of hsa_circ_0092856 in NSCLC were analyzed via qRT-PCR, a CCK-8 assay, a plate cloning experiment, scratch testing, a transwell chamber experiment, an RNA nuclear mass separation experiment, an RIP experiment, and a Western blot test. The results showed that hsa_circ_0092856 was highly expressed in NSCLC cells, and the knockdown of hsa_circ_0092856 could inhibit the proliferation, migration, and invasion of NSCLC cells. The overexpression of hsa_circ_0092856 has the opposite effect. The expression of eIF3a also changed with the change in hsa_circ_0092856. These results suggest that hsa_circ_0092856 may play a key role in the progression of NSCLC by regulating the expression of eIF3a.

## 1. Introduction

Lung cancer is the cancer with the highest incidence and mortality. In 2018, about 2 million new cases of lung cancer and 18,000 deaths were recorded worldwide [1,2]_._ Lung cancer has been divided into two main subsets according to pathology: small-cell lung cancer, accounting for 20% of the total number of cases [3], and non-small-cell lung cancer, accounting for 80% [4]. Non-small-cell lung cancer is divided into three categories: large-cell lung cancer (10%), lung squamous cell carcinoma (30%), and lung adenocarcinoma (40%). The latest progress in diagnosis and treatment procedures has steadily increased the survival rate of most cancers, but the 5-year survival rate of lung cancer is only 17.7% [5]. Compared with patients with advanced non-small-cell lung cancer, the survival rate of patients with early non-small-cell lung cancer is significantly improved. Therefore, it is crucial to establish the molecular mechanism underlying the tumorigenesis of non-small-cell lung cancer, and to identify effective diagnostic biomarkers and therapeutic targets for non-small-cell lung cancer.

Circular RNA (CircRNA) is a class of non-coding transcripts, characterized by a closed-loop structure, without 5′-3′ polarity, which is difficult to be digested by RNase R and more stable than linear RNA. The functions of circRNA include being an miRNA sponge, acting as transcription factors, interacting with RNA binding proteins, and translation into proteins. In recent years, circRNA has become a hot topic [6,7]. Previous studies have indicated that circRNAs are implicated in tumor progression and affect cell proliferation, metastasis, and apoptosis [8]. CircRNA, as a competitive endogenous RNA (ceRNA) mechanism, has been widely studied in various cancer research. CircMBOAT2 promotes prostate cancer progression through the PI3K/Akt pathway, mediated by the miR-1271-5p/mTOR axis [9]. CircNRIP1 silencing inhibits the progression of gastric cancer by up-regulating miR-182 and down-regulating ROCK1 [10]. CircRNA_100859 regulates colon cancer progression through the miR-217-HIF-1α axis [11]; circ_0005394 promotes the growth and invasion of hepatocellular carcinoma by regulating the miR-507/E2F3 and miR-515-5p/CXCL6 signaling pathways [12]; and the down-regulation of Circ-PTN inhibits glioma cell proliferation, invasion, and glycolysis by regulating the miR-432-5p/RAB10 axis [13]. The circular RNA circSATB2 can regulate the expression of FSCN3 by directly binding to miR-1, further promoting the progression of NSCLC, and may be involved in intercellular communication through exosomes [14]. Exosome-transmitted circVMP1 facilitates the progression and cisplatin resistance of non-small-cell lung cancer by targeting the miR-524-5p-METTL3/SOX2 axis [15]. The above studies indicate that circRNAs are closely related to human cancer. However, the role of circRNAs in the occurrence and development of lung cancer needs to be further explored.

Eukaryotic translation initiation factor 3 subunit A (eIF3a) is the largest subunit of the eIF3 family, which is a key point in all steps of translation initiation [16]. Our previous studies showed that the expression of eIF3a protein is markedly associated with the response of lung cancer patients to platinum-based chemotherapy [17]; the genetic polymorphism of *eIF3a rs3740556* was associated with the severe nephrotoxicity and ototoxicity caused by platinum-based chemotherapy in NSCLC patients; and the A/G allele of *eIF3a rs3740556* predicted platinum-based chemotherapy resistance in lung cancer patients [18,19]. Moreover, we found that *eIF3a R803K* (*rs77382849*) mutation mediates chemotherapy resistance by inducing cellular senescence in small-cell lung cancer [20]. Recently, we screened 31 eIF3a-derived circRNAs, in which two circeIF3as were identified to be correlated with cisplatin drug sensitivity in lung cancer [6].

In this study, we screened for biologically functional circRNA hsa_circ_0092856 among the 31 circRNAs derived from *eIF3a.* We evaluated the role of hsa_circ_0092856 in the proliferation, migration, and invasion of NSCLC cells. It was also found that hsa_circ_0092856 played a role in NSCLC cells by regulating the expression of eIF3a. In addition, the role of hsa_circ_0092856 in the progression of NSCLC was analyzed. The results of this study reveal a novel mechanism underlying the progression of lung cancer and propose potential new molecular targets for lung cancer research.

## 2. Materials and Methods

### 2.1. Cell Lines

This study was to investigate the effect of circRNA on the progression of non-small-cell lung cancer. Therefore, we selected a normal pulmonary fibroblast cell line, MRC-5, and NSCLC cells A549, H226, and H1299 as experimental materials. MRC-5, A549, H226, and H1299 cells were purchased from the Chinese Academy of Sciences (Shanghai, China). The MRC-5 cells were cultured in an MEM medium (Gibco, Grand Island, NY, USA) containing 10% fetal bovine serum (Gibco, Grand Island, NY, USA), and the H226, A549, and H1299 cells were cultured in an RPMI-1640 medium (Gibco, Grand Island, NY, USA) containing 10% fetal bovine serum. All cells were cultured in a 5% CO_2_ incubator at 37 °C.

### 2.2. Cell Transfection

To silence hsa_circ_0092856, we purchased hsa_circ_0092856-targeting small interfering RNAs (siRNAs) from RiboBio (Guangzhou, China) and transfected them into cells using the Lipofectamine iMax (Invitrogen, Carlsbad, CA, USA) according to the manufacturer’s instructions. In order to overexpress gene A, we purchased the hsa_circ_0092856 plasmid from Geneseed and transfected it into cells using the Lipofectamine 3000 (Invitrogen, Carlsbad, CA, USA) according to the manufacturer’s instructions. After cell culture for 24 to 48 h, the following experiments were performed.

### 2.3. RNA Isolation and Quantitative Real-Time Polymerase Chain Reaction

Total RNA was extracted using TRIzol (Invitrogen, Carlsbad, CA, USA). The newly extracted RNA was subjected to concentration detection using the Biospec-nano nucleic acid analyzer. We used the cytoplasm and nuclear RNA purification kit (Norgen, Thorold, ON, Canada) to extract cytoplasm and nuclear RNA according to the manufacturer’s instructions. cDNA was synthesized using a Primescript RT kit (Takara, Tokyo, Japan) with a gDNA eraser according to the manufacturer’s instructions. A TB Green Premix Dimer Eraser Assay Kit (Takara, Tokyo, Japan) was used to evaluate circRNA expression via a real-time quantitative PCR with the Roche Light Cycler 480 PCR system (LightCycler® 480, Basel, Switzerland). All primer sequences are presented in Table 1.

### 2.4. RNase R Treatment

Total RNA was extracted using TRIzol (Invitrogen, CA, USA). The newly extracted RNA was subjected to concentration detection using the Biospec-nano nucleic acid analyzer. We used the cytoplasm and nuclear RNA purification kit (Norgen, Canada) to extract cytoplasm and nuclear RNA according to the manufacturer’s instructions. The RNA RNase R treatment was performed using 1 μL of RNase R (20 U/μL) at 37 °C for 10 min. 

### 2.5. Western Blot Analysis

After 48 h of transfection, total protein was extracted using an RIPA lysate (Beyotime Biotechnology, Shanghai, China) supplemented with a protease inhibitor cocktail (Roche, Basel, Switzerland) and a phosphatase inhibitor cocktail (Beyotime Biotechnology, China). The protein concentration was determined through a BCA assay (Bio-Rad, Hercules, CA, USA). The standard method was used for the protein imprinting analysis, and then transferred to PVDF membranes (Millipore, Bedford, MA, USA). After blocking for 2 h, the PVDF membranes were incubated overnight with an anti-mouse β-actin antibody (Sigma-Aldrich, St. Louis, MO, USA) and an anti-rabbit eIF3a antibody (Cell Signaling Technology, Danvers, MA, USA) at 4 °C. We then incubated the PVDF membranes with the corresponding secondary antibody for 1 h. Images were captured using a ChemiDoc System (Tanon 5200 Multi, Beijing, China).

### 2.6. Cell Viability Assay

The treated cells were seeded in five 96-well plates (1000/well), to one of which CCK-8 was added (Bimake, Houston, TX, USA); incubated in a 37 °C 5% CO_2_ incubator for 1.5 h; and then measured at 450 nm by using an Eon plate reader, Then, we compared the cell quantity based on the magnitude of the absorbance values to detect cell viability. The rest were also cultured in an incubator, and CCK-8 was added every 24 h to detect one piece.

### 2.7. Colony Formation Assay

We inoculated the pre-treated cells into a six-well plate, inoculating 1000 cells per well. After culturing for 10–15 days, we washed both sides with PBS, fixed the cells with 4% paraformaldehyde at room temperature for 15–30 min, and then stained them with crystal violet. Finally, we took pictures and counted the number of cells.

### 2.8. RNA Fluorescence In Situ Hybridization (FISH)

The specific probe of hsa_circ_0092856 was designed and provided by Axl-bio (Guangzhou, China), and the sequence is shown in Table 2. An RNA FISH assay was performed using a FISH detection kit (Axl-bio, Guangzhou, China) according to the manufacturer’s instructions. An image analysis was performed with a laser scanning confocal microscope (LSM800, Carl Zeiss, Jena, Germany).

### 2.9. RNA Immunoprecipitation Assay

An RNA immunoprecipitation assay (RIP) was performed using an EZMagna RIP Kit (Millipore, Billerica, MA, USA) according to the manufacturer’s instructions. A549 cell lysates were incubated with anti-rabbit AGO_2_ antibodies (Abcam, Cambridge, MA, USA) or anti-rabbit IgG antibodies (Millipore, Billerica, MA, USA).

### 2.10. Wound-Healing Experiment

We inoculate an appropriate amount of cells into a six-well plate, after the cells had grown to a suitable density. We used a marker to draw a horizontal line on the back of the six-well plate. Transfection was performed using the same method. Once the cells had grown to full confluency, the wound was scratched and washed 3 times with PBS, and a serum-free culture was added. Pictures were taken at 0 h and 48 h; finally, we measured the width of the scratch and evaluated the healing condition.

### 2.11. Invasion and Migration Experiments 

In the invasion experiment, the transfected cells were seeded into a cell pre-added with Matrigel to 100,000 cells per well. In the migration experiment, the transfected cells were seeded into a cell with 50,000 cells per well. We removed cells from upper chamber after 24 h of culture incubation, fixed the cells with 4% paraformaldehyde at room temperature for 15–30 min, stained them with crystal violet, and finally photographed and counted them.

### 2.12. Statistical Analysis

In this study, SPSS 19.0 (SPSS Inc., Chicago, IL, USA) and GraphPad Prism 8.0 (GraphPad Inc., San Diego, CA, USA) were used for statistical analysis and plotting. For the measurement data, Student’s *t* test was used for comparisons between two groups, and a one-way ANOVA test was used for comparisons between multiple groups. The data in this study are expressed as the mean ± standard deviation, which is considered to have significant differences when *p* < 0.05.

## 3. Results

### 3.1. Hsa_circ_0092856 Was Highly Expressed in NSCLC Cells

Our previous studies found that eIF3a is highly expressed in lung cancer and promotes the proliferation, migration, and invasion of lung cancer cells [21]. All of us conjecture that circRNA derived from eIF3a may also have important regulatory functions in lung cancer. We found 31 circRNAs derived from *eIF3a* in the Circbase database, including eight circRNAs (EcircRNAs) formed of exons and 23 circRNAs (EIcircRNAs) formed of exon–introns (Table 3).

In this study, the RT-PCR results showed that 10 of the 31 CircRNAs were expressed in MRC-5 and A549 cells. We selected one circRNA formed of exons with the largest difference in expression, which was called hsa_circ_0092856 (Figure 1A). The genomic location of hsa_circ_0092856 is shown in Figure 1B. Hsa_circ_0092856 was derived from the back-splicing of exon 6 of its host gene, *eIF3a* (Figure 1B). We detected the expression of hsa_circ_0092856 and Eif3a in MCR-5, A549, H1299, and H226 cells through the RT-qPCR; the results showed that the expression of hsa_circ_0092856 and *eIF3a* in lung cancer cell lines A549, H1299, and H226 are significantly up-regulated compared with that in the normal human lung fibroblast cell line, MRC-5 (Figure 1C). In addition, we detected the stability of hsa_circ_0092856 in H1299 cells and found that hsa_circ_0092856 is resistant to RNase R+ digestion compared with *eIF3a* mRNA (Figure 1D). The above results indicate that hsa_circ_0092856 is highly expressed in NSCLC cells, and may influence some biological functions of NSCLC cells. 

### 3.2. Hsa_circ_0092856 Facilitated the Proliferation, Migration, and Invasion of NSCLC Cells

Based on the results from Figure 1, we conducted functional experiments of hsa_circ_0092856 by knocking down and overexpressing it in H1299 and A549 cell lines, respectively; hsa_circ_0092856 can be significantly knocked down or overexpressed (Figure 2A). The effects of hsa_circ_0092856 on the proliferation, migration, and invasion of NSCLC cells were analyzed in a CCK-8 experiment, a colony formation experiment, a wound-healing experiment, and a transwell chamber experiment. The CCK-8 and colony formation assay showed that hsa_circ_0092856 knockdown inhibited the growth and colony formation of H1299 cells, while hsa_circ_0092856 overexpression promoted the growth and colony formation of A549 cells (Figure 2B,C). The wound-healing test and transwell chamber test showed that hsa_circ_0092856 knockdown could inhibit the migration of H1299 cells, while hsa_circ_0092856 overexpression could promote the migration of A549 cells (Figure 2D,E). The transwell chamber test showed that hsa_circ_0092856 knockdown could inhibit the invasion of H1299 cells, while hsa_circ_0092856 overexpression could promote the invasion of A549 cells (Figure 2F). The Western blot test showed that has_circ_0092856 knockdown could inhibit the protein level of p-Akt and bcl2 and elevate the protein level of BAX and e-cadherin; has_circ_0092856 overexpression is the opposite. The above results showed that hsa_circ_0092856 enhanced the proliferation, migration, and invasion of NSCLC cells.

### 3.3. The Biological Functions of Hsa_circ_0092856

CircRNAs may have different biological functions due to their different localization and distribution in cells. For example, their functions include miRNA sponging, interaction with transcription factors, RNA-binding protein interactions, and translation into proteins. Most circRNAs are located in the cytoplasm and generally have the function of being an miRNA molecular sponge. In order to speculate the biological function of hsa_circ_0092856, we performed RNA fluorescence in situ hybridization (FISH) in H1299 cells to determine the subcellular localization of hsa_circ_0092856. In the present study, the FISH results showed that hsa_circ_0092856 was localized in the cytoplasm (Figure 3A), which implicated that hsa_circ_0092856 may have the function of being an miRNA molecular sponge. The RIP experiment results showed that AGO2 magnetic beads could not enrich hsa_circ_0092856 more compared with IgG magnetic beads in H299 cells (Figure 3B), indicating that hsa_circ_0092856 may not have an miRNA sponge function. CircRNA has been proven to regulate gene expression in a variety of ways. For example, ciRNAs such as ci-ankrd52 and ci-sirt7 can promote the transcription of their parent genes by interacting with RNA Pol II complexes [22,23]. Furthermore, circBRD7 enhances the transcriptional activity and expression of its host gene, *BRD7*. As a tumor suppressor, circBRD7 inhibits cell proliferation, migration, and invasion, as well as the growth and metastasis of xenograft tumors [24]. The circular RNA circITGA7, formed by exon 4, regulates the Ras pathway and up-regulates the transcriptional repression of its host gene, *ITGA7*, thereby inhibiting the growth and metastasis of colorectal cancer [25]. Therefore, we speculated that hsa_circ_0092856 may play a biological function by regulating the expression of its parent gene, *eIF3a*. We knocked down and over-expressed hsa_circ_0092856 in H1299 and A549 cell lines, respectively. The RT-qPCR and Western blot results showed that the mRNA and protein levels of the parental gene *eIF3a* in the hsa_circ_0092856 knockdown group were significantly lower than those in the control group (Figure 3C,D). After the overexpression of hsa_circ_0092856, the mRNA and protein levels of eIF3a were significantly higher than those of the control group (Figure 3C,D). The above results show that hsa_circ_0092856 could play a biological role by regulating the expression of parental gene *eIF3a*.

## 4. Discussion

Andreeva and Cooper reported in 2015 that circRNA widely exists in animal and plant cell tissues and has a variety of specific biological characteristics, which attracted the attention of many researchers [26]. With the development of biological sequencing technology, more and more circRNAs have been found; circRNAs have become a research hotspot from their original junk RNA. A large number of studies have found that circRNAs play an important role in various human diseases [6]. In addition, circRNA is also a promising cancer treatment target, including in NSCLC [27].

As the largest subtype of eIF3, eIF3a plays a key role in translation initiation. Over the past few years, *eIF3a* has been identified as a proto-oncogene and is widely reported to be associated with tumorigenesis, metastasis, prognosis, and treatment response [16,28,29,30]. More and more studies have reported the overexpression of eIF3a in several cancers. The ectopic up-regulation of eIF3a promotes cell proliferation and malignant transformation [20]. eIF3a sustains non-small-cell lung cancer stem cell-like properties by promoting the YY1-mediated transcriptional activation of β-catenin [21]. miR-875-5p is down-regulated in hepatocellular carcinoma, and it inhibits tumor growth and metastasis by targeting the 3’UTR of *eIF3a* to down-regulate its expression [31].

Recent studies have also found that some CircRNAs can regulate the occurrence, development, and drug sensitivity in non-small-cell lung cancer. For example, the down-regulation of hsa_circ_0007580 can inhibit the tumorigenesis of non-small-cell lung cancer by reducing the expression of miR-545-3p [32]. hsa_circ_0002483 inhibits the progress of non-small-cell lung cancer and improves the sensitivity of paclitaxel by targeting miR-182-5p [33]. Circular RNA circHMGB2 drives immune suppression and anti-PD-1 resistance in lung adenocarcinoma and squamous cell carcinoma through the miR-1a-181p/CARM5 axis [34]. Circular RNA_100565 promotes cisplatin resistance in NSCLC cells by regulating proliferation, apoptosis, and autophagy through the miR-337-3p/ADAM28 axis [35]. Circular RNA hsa_circ_0070659 predicts poor prognosis and promotes the progression of non-small-cell lung cancer through the miR-377/Ras-related protein 3C pathway [36]. Some studies have shown that some genes and their generated circRNA have similar biological functions, so we speculated that circRNA derived from the *eIF3a* gene may also have similar biological functions with to parent gene in non-small-cell lung cancer. We identified 31 circRNAs derived from *eIF3a* in the database and selected hsa_circ_0092856 for further investigation in A549 and H1299 cells. In the current study, we found that hsa_circ_0092856 was highly expressed in NSCLC cell lines, and that the knockdown of hsa_circ_0092856 inhibited the proliferation, migration, and invasion of H1299 and A549 cells; however, the overexpression of hsa_circ_0092856 promoted the proliferation, migration, and invasion of H1299 and A549 cells, which played this biological function by regulating the expression of eIF3a. Although there are many studies on circRNA in NSCLC, the biological function of circRNA derived from eIF3a in regulating parental gene expression has been discovered first, which provides a new direction for the study of circRNA in NSCLC. However, the molecular mechanism underlying its biological functions remains to be investigated. There is still a lack of in vivo research on animal models and clinical studies, and further in-depth studies are needed. We believe that through an in-depth investigation of the molecular mechanisms, new research directions will be provided for the diagnosis and treatment of non-small-cell lung cancer.

In summary, our study found that hsa_circ_0092856 promoted the proliferation, migration, and invasion of NSCLC cell lines by up-regulating the expression of parental gene *eIF3a*, which explained how hsa_circ_0092856 can play an oncogene role in NSCLC. This study provides a new research direction for the early diagnosis and treatment of non-small-cell lung cancer.

## Figures and Tables

**Figure 1 biomedicines-12-00247-f001:**
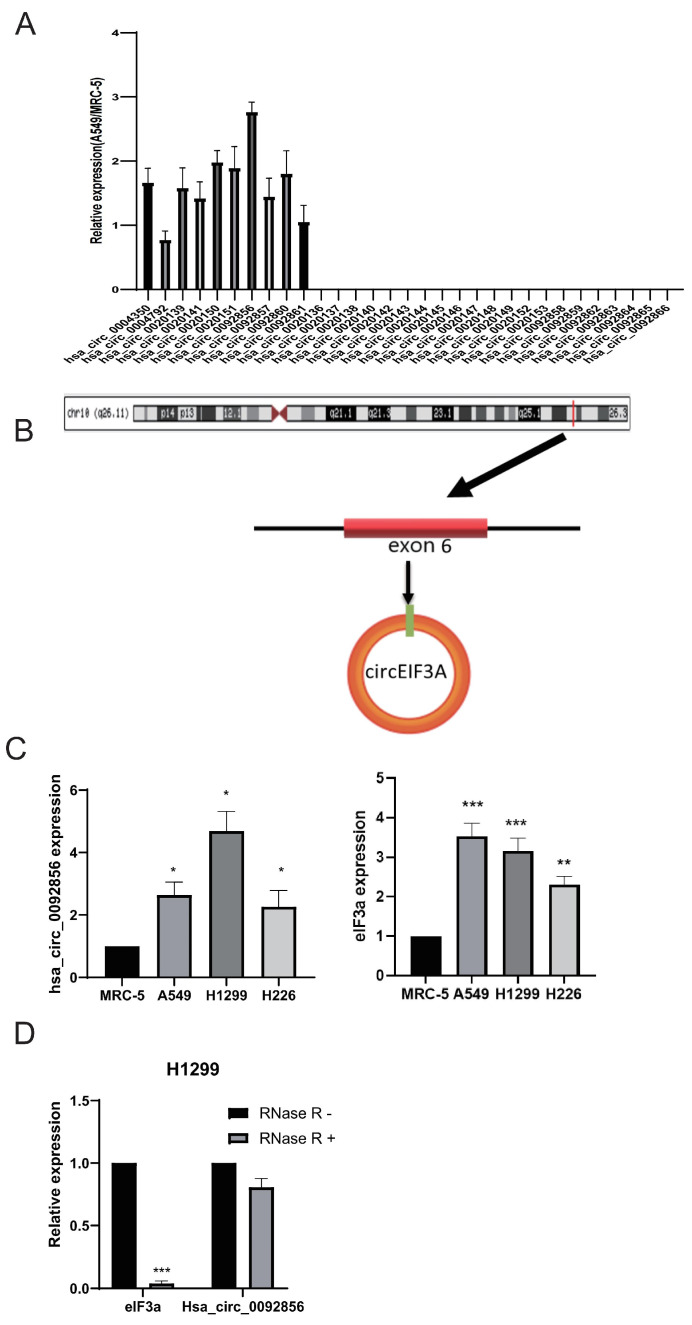
Hsa_circ_0092856 was highly expressed in NSCLC cells. (**A**) Compared with MRC-5 cells, the expression of 31 CircRNAs in A549 cells was different. β-actin is a reference gene; *n* = 3. (**B**) Map of genome location and splicing pattern of Hsa_circ_0092856. (**C**) The expression levels of Hsa_circ_0092856 and *eIF3a* in MRC-5, A549, H1299 and H226 cell lines. β-actin was an internal reference gene, * *p* < 0.05, ** *p* < 0.01 and *** *p* < 0.001 compared with the MRC-5 group; *n* = 3. (**D**) After RNase R treatment, the expression levels of *eIF3a* and hsa_circ_0092856s were determined via qRT-PCR. β-Actin was an internal reference gene. *** *p* < 0.001 compared with RNase R-group; *n* = 3.

**Figure 2 biomedicines-12-00247-f002:**
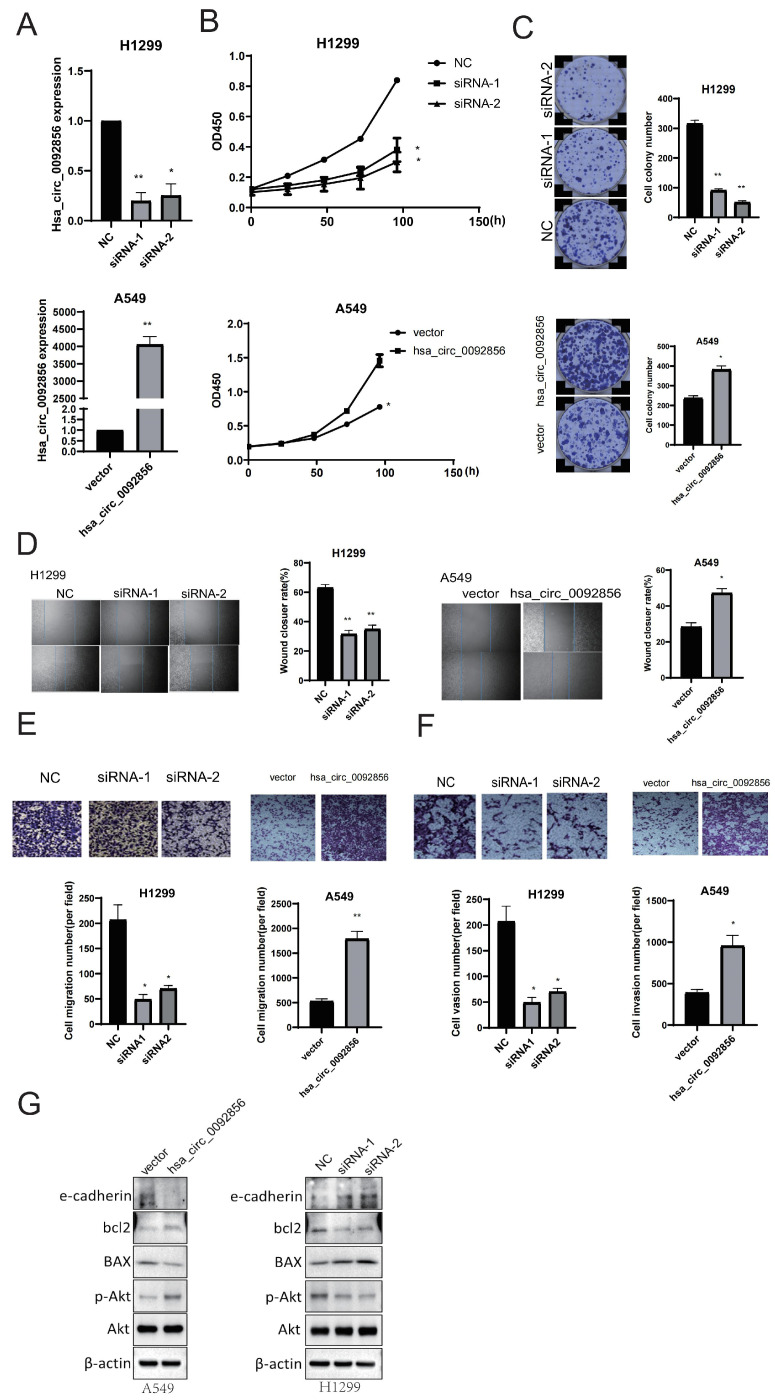
Has_circ_0092856 facilitated proliferation, migration, and invasion of NSCLC cells. (**A**) Knockdown efficiency of siRNA vector and overexpression efficiency of plasmid of hsa_circ_0092856. β-Actin was an internal reference gene. * *p* < 0.05, ** *p* < 0.001 compared with the NC group; *n* = 3. (**B**,**C**) Effect of hsa_circ_0092856 knockdown and overexpression on proliferation of NSCLC cells via CCK-8 assay and colony formation assay; * *p* < 0.05, ** *p* < 0.01 compared with NC group; *n* = 3. (**D**,**E**) Effect of hsa_circ_0092856 knockdown and overexpression on migration of NSCLC cells via transwell assay and wound-healing assay; * *p* < 0.05, ** *p* < 0.01 compared with NC group; *n* = 3. (**F**) Effect of *hsa_circ_0092856* knockdown and overexpression on invasion of NSCLC cells via transwell assay; * *p* < 0.05, ** *p* < 0.01 compared with NC group; *n* = 3. (**G**) Effect of *hsa_circ_0092856* knockdown and overexpression on proliferation, migration, and invasion markers of NSCLC cells, with β-actin as a reference gene.

**Figure 3 biomedicines-12-00247-f003:**
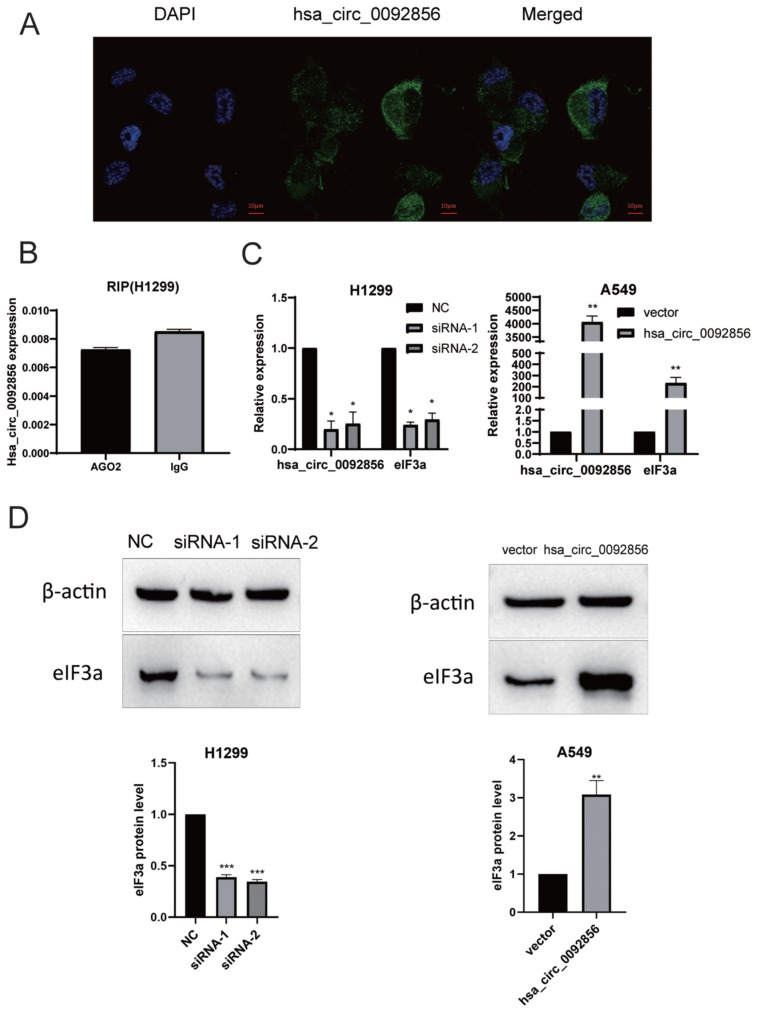
Possible biological functions of has_circ_0092856. (**A**) The subcellular localization of hsa_circ_0092856 was detected via RNA fluorescence in situ hybridization (FISH). (**B**) Verification of miRNA sponge function of hsa_circ_0092856 via RIP. (**C**,**D**) The hsa_circ_0092856 was knocked down and overexpressed in the two cell lines, respectively, and the changes in the mRNA and protein levels of the parental gene eIF3a were detected. β-actin was a reference gene; * *p* < 0.05, ** *p* < 0.01 and *** *p* < 0.001 compared with NC group; *n* = 3.

**Table 1 biomedicines-12-00247-t001:** Primer sequences used for qRT-PCR.

RNA ID	Primer Sequences
hsa_circ_0004350	F: 5′-TGGGAGTCTTACAGGCAGTG-3′ R: 5′-ACATCCAGAGCAGGCTGCTT-3′
hsa_circ_0004792	F: 5′-CCTTGATTCTCGAGGTGGAC-3′ R: 5′-GGAGGAGAATGACAAGGACC-3′
hsa_circ_0020136	F: 5′-CTCATTGAAGCCAACACAGAAC-3′ R: 5′-ATGGACTGGCTCTCTGGATTAT-3′
hsa_circ_0020137	F: 5′-TAAGAGACGACAGGGACCGAAG-3′ R: 5′-AGAAGGCGGAGGATGGTGTT-3′
hsa_circ_0020138	F: 5′-TCGGGAGGAGAATGACAAGG-3′ R: 5′-AAGGCGGAGGATGGTGTTGT-3′
hsa_circ_0020139	F: 5′-GAGAAGACTTGGCGATAGTTCC-3′ R: 5′-GGACCAATCGGAGCATCTT-3′
hsa_circ_0020140	F: 5′-GGCTCTTGAACATAAGAATCG-3′ R: 5′-TCTTCACCACTTACAGCACTTAG-3′
hsa_circ_0020141	F: 5′-AGCTGCACGGCAGTCTGTTTAT-3′ R: 5′-CGAAGATCCACGCAAAGTTCC-3′
hsa_circ_0020142	F: 5′-ACAGCCATGTCCTCAGTACTTG-3′ R: 5′-AGAACCTTTGTGACTCGCTCAC-3′
hsa_circ_0020143	F: 5′-GTCAAACGAGAAAACTCAATGC-3′ R: 5′-CGTTTGACTTCTTTGGTTCCTT-3′
hsa_circ_0020144	F: 5′-CAACTGGAACGGGCCATAG-3′ R: 5′-AGACGAAGGGGAACCAGC-3′
hsa_circ_0020145	F: 5′-CTGCAAAACAACACCATCCTC-3′ R: 5′-GGTTCCTTTTCAGGTTGTTCC-3′
hsa_circ_0020146	F: 5′-AACAACACCATCCTCCGCCT-3′ R: 5′-TGCTGCAATTCCGGTTCCT-3′
hsa_circ_0020147	F: 5′-GAACAACCTGAAAAGGAACCG-3′ R: 5′-GCAGCTTCAGTTTTTTCCTCTG-3′
hsa_circ_0020148	F: 5′-ACCGACACGAATTGGCCT-3′ R: 5′-CGAGCAATATCCGTACGCTC-3′
hsa_circ_0020149	F: 5′-TGCAACACTACTAGGTCTTCAAGC-3′ R: 5′-CTTCAGTTTTTTCCTCTGCCAT-3′
hsa_circ_0020150	F: 5′-ACTACTAGGTCTTCAAGCCCCAC-3′ R: 5′-AAGTGGCTCTTGCGAAGATC-3′
hsa_circ_0020151	F: 5′-GCATCTACACTCCATCGTCTTTAC-3′ R: 5′-TTACGGAATTCAGCCTTACGC-3′
hsa_circ_0020152	F: 5′-GGTTAAATTCCTGTGGGAGTCT-3′ R: 5′-TGTCTTCACCACTTACAGCACTT-3′
hsa_circ_0020153	F: 5′-TCTTACAGGCAGTGTTTGGACC-3′ R: 5′-ATGCCCTAACAACATCCTCCAG-3′
hsa_circ_0092856	F: 5′-AGGCGCTGATGATGAGCGAT-3′ R: 5′-TTCCTCAGGACCACGTCTAG-3′
hsa_circ_0092857	F: 5′-AACGAGAACGGCGTAGAGAGG-3′ R: 5′-TAAACAGACTGCCGTGCAGC-3′
hsa_circ_0092858	F: 5′-GTAATGCGACTCAAAGCTGCAC-3′ R: 5′-TCTCTGTTCCTCGTAAGCGCTC-3′
hsa_circ_0092859	F: 5′-GTAATGCGACTCAAAGCTGCAC-3′ R: 5′-GAGGACCAATCGGAGCATCTT-3′
hsa_circ_0092860	F: 5′-AGACATGGATCTGTGGGAGC-3′ R: 5′-TCTCTGTTCCTCGTAAGCGC-3′
hsa_circ_0092861	F: 5′-TGGCTAAACAGGTTGAACAAC-3′ R: 5′-TGTTCAACCTGTTTAGCCATG-3′
hsa_circ_0092862	F: 5′-ACAGCCATGTCCTCAGTACTTG-3′ R: 5′-GAATTCAGCCTTACGCGTG-3′
hsa_circ_0092863	F: 5′-TGCATCTACACTCCATCGTCTT-3′ R: 5′-TCCACGCAAAGTTCCAAGTAT-3′
hsa_circ_0092864	F: 5′-CCATGCATTTGGAAACCAGACT 3′ R: 5′-CTCCCACAGGAATTTAACCCAT 3′
hsa_circ_0092865	F: 5′-TGAGAGTGTTCTCCTAAGTGCTGT-3′ R: 5′-GCTGAGATTCTTCTTTAGCAGCT-3′
hsa_circ_0092866	F: 5′-GCTGCTAAAGAAGAATCTCAGCAG-3′ R: 5′-CCACGCAAAGTTCCAAGTATTTC-3′
eIF3a	F: 5′-TCAAGTCGCCGGGACGATA-3′ R: 5′-CCTGTCATCAGCACGTCTCCA-3′

**Table 2 biomedicines-12-00247-t002:** FISH probes sequences of hsa_circ_0092856.

Type	Sequences
FISH probes	TATCTTCCTCAGGACCACGTCTAGGGCCCCGGTCATCATCCAT

**Table 3 biomedicines-12-00247-t003:** CircRNA screening from *eIF3a* gene.

CircRNA Type	Total	CircRNA ID		
EcircRNAs	8	hsa_circ_0004792	hsa_circ_0020143	hsa_circ_0020145
		hsa_circ_0020148	hsa_circ_0020152	hsa_circ_0092856
		hsa_circ_0092860	hsa_circ_0092861	
EIcircRNAs	23	hsa_circ_0004350	hsa_circ_0020136	hsa_circ_0020137
		hsa_circ_0020138	hsa_circ_0020139	hsa_circ_0020140
		hsa_circ_0020141	hsa_circ_0020142	hsa_circ_0020144
		hsa_circ_0020146	hsa_circ_0020147	hsa_circ_0020149
		hsa_circ_0020150	hsa_circ_0020151	hsa_circ_0020153
		hsa_circ_0092857	hsa_circ_0092858	hsa_circ_0092859
		hsa_circ_0092862	hsa_circ_0092863	hsa_circ_0092864
		hsa_circ_0092865	hsa_circ_0092866	

## Data Availability

All other data are included within the article or available from the authors on request.

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
