# Peer review of "Hsa_circ_0092856 Promoted the Proliferation, Migration, and Invasion of NSCLC Cells by Up-Regulating the Expression of eIF3a"

_biomedicines, 2024, doi:10.3390/biomedicines12010247_

Round 1
Reviewer 1 Report
Comments and Suggestions for Authors
Dear Authors,
Biomedicines-2662735
Hsa_circ_0092856 Promoted the Proliferation, Migration, and 2 Invasion of NSCLC Cells by Up-Regulating the Expression of 3 eIF3a by Yuan et al describes the role of circular RNA in a variety of human 24 malignancies such as non-small cell lung cancer (NSCLC). In current study, the authors explored the role of 25 hsa_circ_0092856 in the progression of NSCLC. The authors screened CircRNA from eIF3a gene in Circbase 26 database. The biological functions of hsa_circ_0092856 in NSCLC were analyzed by qRT-PCR, CCK- 8 assay, plate cloning experiment, scratch test, transwell chamber experiment, RNA nuclear mass 28 separation experiment, RIP experiment and western blot test. The results showed that 29 hsa_circ_0092856 was highly expressed in NSCLC cells, and the knock-down of hsa_circ_0092856 could inhibit the proliferation, migration and invasion of NSCLC cells. The overexpression of hsa_circ_0092856 has the opposite effect. The expression of eIF3a also changed with the change hsa_circ_0092856. These results suggest that hsa_circ_0092856 may play a key role in the progression of NSCLC by regulating the expression of eIF3a.
1. The article is interesting, however the authors could show proliferation in in vivo with over expression of circ_0092856 and no proliferation with the silencing of circ_0092856.
2. Flow cytometry with over expression and silencing with circ_0092856. Will improve the manuscript quality.
Comments on the Quality of English LanguageProof reading required
Author Response
Dear reviewer,Thank you very much for your valuable comments.
1.Our work has shown that knockdown and overexpression of hsa_circ_0092856 have an effect on cell proliferation, as shown in figure 2B and figure 2C.
2.Due to the lack of a flow cytometer in our facility, we need to apply to another institution for its use. In addition, reagents also need to be ordered, and these work can not be completed within the ten days of repair. However, we added WB experiments on the effects of overexpression and knockdown of hsa_circ_0092856 on proliferation and apoptosis to confirm our experimental results.
Reviewer 2 Report
Comments and Suggestions for Authors
Dear authors,
I read your article and I find the research on cicrRNA quite intriguing. The field holds immense potential for various applications. However, I would like to offer some constructive feedback and suggestions for improvement:
1. I recommend revising the wording on Line 94 to avoid potential confusion. The current phrasing might lead readers to believe that all cell lines are derived from normal pulmonary fibroblasts. Please consider rephrasing to provide a clearer understanding of the diverse origins of the cell lines.
2. I noticed that there is no mention of the use of mitomycin in your wound healing experiments. It could strengthen your argument about the observed increase in cell migration and metastatic dynamics. It would be beneficial if you could provide an explanation for the omission of mitomycin and discuss its potential impact on your results.
3. I recommend enlarging the images in Figure 2 or considering a different format to enhance clarity. The current format appears a bit confusing, and larger images would aid readers in better understanding the details presented.
Author Response
1.Dear reviewer,According to your suggestions, revisions have been made in the manuscript.
2.Dear reviewer,Cytosporone is a drug that inhibits cell division, thereby resulting in the suppression of cell migration. In our experiments, scratch assays were performed using serum-free culture medium, which allows us to disregard the influence of cell proliferation. This information has been added to the methodology section in the main text.
3.Dear reviewer,Thank you very much for your valuable comments.The image has been replaced with a clearer one.
Reviewer 3 Report
Comments and Suggestions for Authors
Major comments:
1. Please cite the reference about the description in section 3.1: “Our previous studies found that eIF3a is highly expressed in lung cancer and promotes the proliferation, migration, and invasion of lung cancer cells.”
2. Migration and invasion markers can be detected in siRNA-H1299 and circ_0092856-A549.
3. eIF3a can be detected among the cell lines shown in Fig 1C.
4. What is the mechanism to mediate eIF3a expression by circ_0092856.
5. If circ_0092856 majorly locates in cytoplasm (Fig 3A), how does it interact with RNA Pol II complexes or to determine gene expression.
Author Response
1.Dear reviewer,Thank you very much for your valuable comments.The referenced literature has been cited in the manuscript.
2.Dear reviewer, according to your suggestions, the experiment has been added in figure 2G.
3.Dear reviewer, according to your suggestions, the experiment has been added in figure 1C.
4.Dear reviewer, our research group's previous work mainly focused on eIF3a, and now we find that hsa_circ_0092856 transcribed from eIF3a also has biological functions. The specific mechanism of hsa_circ_0092856 mediating eIF3a expression is the future research direction of this paper.
5.Dear reviewer,The referenced literature suggests that the circITGA7 located in the cytoplasm can inhibit the transcription factor RREB1 via the Ras pathway, thereby promoting the transcription of its host gene ITGA7. In this study, hsa_circ_0092856 can also affect the expression of its host gene through other mechanisms besides interacting with RNA Pol II complex. We have added the answer to your question in the text.
Round 2
Reviewer 1 Report
Comments and Suggestions for Authors
The manuscript is improved
Reviewer 3 Report
Comments and Suggestions for Authors
no more questions.